# Body mass index trends and its impact of under and overweight on outcome among PLHIV on antiretroviral treatment in rural Tanzania: A prospective cohort study

**Aneth Vedastus Kalinjuma**[1,2☯]*, **Hannah Hussey**[1,3☯], **Getrud Joseph Mollel**[1,4],
**Emilio Letang**[5,6], **Manuel Battegay**[7], **Tracy R. Glass**[5,8], **Daniel Paris**[5,8],
**Fiona Vanobberghen**[5,8‡], **Maja Weisser**[1,5,7,8‡], **on behalf of the KIULARCO study group**[¶]

1 Department of Interventions and Clinical Trials, Ifakara Health Institute, Ifakara, Tanzania, 2 Department of Epidemiology and Biostatistics, School of Public Health, Faculty of Health Sciences, University of the Witwatersrand, Johannesburg, South Africa, 3 Division of Public Health, School of Public Health and Family Medicine, University of Cape Town, Cape Town, South Africa, 4 St. Francis Referral Hospital, Ifakara, Tanzania, 5 Swiss Tropical and Public Health Institute, Allschwil, Switzerland, 6 Barcelona Institute for Global Health, Hospital Clínic-University of Barcelona, Barcelona, Spain, 7 Division of Infectious Diseases and Hospital Epidemiology, University Hospital Basel, University of Basel, Basel, Switzerland, 8 University of Basel, Basel, Switzerland

☯ These authors contributed equally to this work.
‡ FV and MW also contributed equally to this work.
¶ Membership of the KIULARCO study group is provided in the Acknowledgments.
* avedastus@ihi.or.tz

## Abstract

### Introduction

Increased body weight is an important risk factor for cardiovascular disease and is increasingly reported as a health problem in people living with HIV (PLHIV). There is limited data from rural sub-Saharan Africa, where malnutrition usually presents with both over- and undernutrition. We aimed to determine the prevalence and risk factors of underweight and overweight/obesity in PLHIV enrolled in a cohort in rural Tanzania before the introduction of integrase inhibitors.

### Methods

This nested study of the prospective Kilombero and Ulanga Antiretroviral Cohort included adults aged ≥19 years initiated on antiretroviral therapy between 01/2013 and 12/2018 with follow-up through 06/2019. Body Mass Index (BMI) was classified as underweight (<18.5 kg/m²), normal (18.5–24.9 kg/m²), or overweight/obese (≥25.0 kg/m²). Stratified piecewise linear mixed models were used to assess the association between baseline characteristics and follow-up BMI. Cox proportional hazard models were used to assess the association between time-updated BMI and death/loss to follow-up (LTFU).

**Data Availability Statement:** All datasets used for this manuscript is uploaded in Zenodo. The DOI is 10.5281/zenodo.7699707 The URL is: https://doi.org/10.5281/zenodo.7699707.

**Funding:** This work was supported through the CDCI by the Ministry of Health, Community Development, Gender, Elderly and Children Tanzania; the Government of the Canton of Basel, Switzerland; the Swiss Tropical and Public Health Institute, Switzerland; the University Hospital Basel, Switzerland; the Ifakara Health Institute, Tanzania; and USAID Boresha Afya (through the United States Agency for International Development (USAID) from the President's Emergency Plan for AIDS Relief (PEPFAR) programme). Aneth Vedastus Kalinjuma was supported by the Consortium for Advanced Research Training in Africa (CARTA). CARTA is jointly led by the African Population and Health Research Center and the University of the Witwatersrand and funded by the Carnegie Corporation of New York (Grant No. G-19-57145), Sida (Grant No:54100113), Uppsala Monitoring Center, Norwegian Agency for Development Cooperation (Norad), and by the Wellcome Trust [reference no. 107768/Z/15/Z] and the UK Foreign, Commonwealth & Development Office, with support from the Developing Excellence in Leadership, Training and Science in Africa (DELTAS Africa) programme. The statements made and views expressed are solely the responsibility of the Fellow. The funders had no role in study design, data collection, and analysis, decision to publish, or preparation of the manuscript.

**Competing interests:** I have read the journal's policy and the author of this manuscript has the following competing interests: Emilio Letang is a full-time employee ViiV Healthcare since May 2021. This does not alter our adherence to PLOS ONE policies on sharing data and materials. Other authors have declared that no competing interests exist.

## Results

Among 2,129 patients, 22,027 BMI measurements (median 9 measurements: interquartile range 5–15) were analysed. At baseline, 398 (19%) patients were underweight and 356 (17%) were overweight/obese. The majority of patients were female (n = 1249; 59%), and aged 35–44 years (779; 37%). During the first 9 months, for every three additional months on antiretroviral therapy, BMI increased by 2% (95% confidence interval 1–2%, p<0.0001) among patients underweight at baseline and by 0.7% (0.5–0.6%, p<0.0001) among participants with normal BMI. Over a median of 20 months of follow-up, 107 (5%) patients died and 592 (28%) were LTFU. Being underweight was associated with >2 times the hazard of death/LTFU compared to participants with normal BMI.

## Conclusion

We found a double burden of malnutrition, with underweight being an independent predictor of mortality. Monitoring and measures to address both states of malnutrition among PLHIV should be integrated into routine HIV care.

## Introduction

Increased body weight is an important risk factor for cardiovascular disease [1–3]. Worldwide, the prevalence of overweight (body mass index (BMI) of 25–29 kg/m$^2$) and obesity (BMI ≥30 kg/m$^2$) has steadily increased in both high and low to middle-income countries [1, 4]. Obesity has been regarded as a problem in urban areas, but recent evidence shows that rural areas are increasingly affected too [1]. In addition, low to middle-income countries often have a double burden of malnutrition, that is co-existence of undernutrition and overnutrition in the same population [5].

In people living with HIV (PLHIV), the interaction between HIV and body weight is complex, with low BMI being common in patients with untreated HIV infection and opportunistic infections [6, 7]. Following initiation of antiretroviral treatment (ART), BMI usually increases in parallel to CD4 cell counts [7] and BMI is a predictor of CD4 cell count increase [8, 9]. For underweight PLHIV, the increase in BMI with ART is beneficial, as low BMI is a risk factor for mortality [9–12]. On the other hand, concerns have been raised that the roll-out of ART will "unmask" an obesity epidemic, jeopardizing the improved life expectancy gained from successful ART [13, 14]. Weight gain has been well described with integrase inhibitors such as dolutegravir [15]. However, in a rural African setting, there are limited data on BMI trends on ART before the rollout of integrase inhibitors. Therefore, setting a baseline for studies investigating BMI changes in PLHIV on integrase inhibitors is crucial.

The objectives of this study were to determine the prevalence of underweight and overweight/obesity, trends of BMI, factors associated with follow-up BMI, and the association between time-updated BMI and death/loss to follow-up (LTFU) among adults initiated on ART in rural Tanzania.

## Methods

### Study design, setting, and population

This is a study analyzing prospectively collected longitudinal data captured within the Kilombero and Ulanga Antiretroviral Cohort (KIULARCO). KIULARCO is a cohort study utilizing routine data collected at the Chronic Disease Clinic of Ifakara (CDCI), the HIV care and

treatment center of the St. Francis Referral Hospital in Ifakara, Morogoro region in rural South-Western Tanzania since 2004. CDCI provides care for PLHIV from the surrounding of Kilombero, Ulanga, and Malinyi districts, whose major livelihood is rice farming and fishing. All PLHIV attending the CDCI were asked to participate in KIULARCO and sign an informed consent. Details of KIULARCO have been described elsewhere [16, 17]. In brief, consenting HIV-positive patients are enrolled and followed up monthly in the first 3 months after ART initiation. Thereafter, visits take place 3-monthly–twice yearly by a clinician and twice by a nurse. At every visit, demographic and clinical data are captured, including BMI. Laboratory parameters are measured once to twice yearly.

The study used a database exported in January 2020, and we included KIULARCO participants aged ≥19 years (for adult BMI calculation), who were ART-naïve at recruitment and initiated on ART at the clinic between January 2013-December 2018. We assessed follow-up until June 2019. We excluded transit participants (participants seen for drug refill only), those with prior exposure to ART or who never started ART, those without BMI measurements at ART initiation, and female participants who were pregnant at enrolment or during follow-up. Participants without any follow-up BMI measurements were excluded from trend analyses. The baseline was defined as the date of ART initiation.

## Outcomes and baseline covariates

The primary outcome was BMI, calculated as body weight in kilograms divided by height in squared meters [18]. BMI was classified into five categories according to the World Health Organization (WHO): moderate and severe underweight ($<17.0$ kg/m$^2$), mild underweight ($17.00$–$18.49$ kg/m$^2$), normal BMI ($18.5$–$24.9$ kg/m$^2$), overweight ($25.0$–$29.9$ kg/m$^2$), and obese ($\geq30.0$ kg/m$^2$) [18]. The secondary outcome was time to all-cause mortality or LTFU. LTFU was defined as being $>60$ days late for a scheduled appointment [19], whereby appointments were scheduled three monthly.

Baseline variables were sex, age, education, occupation, CD4 cell count, HIV clinical stage (defined as WHO stage I-IV [20]), and co-morbidities such as arterial hypertension (systolic blood pressure ≥140 mmHg and/or diastolic blood pressure ≥90 mmHg on two consecutive clinic visits), anemia (hemoglobin $<12.9$ g/dL for males aged 19–59 years, $<12.7$ g/dL for males aged ≥60 years, and $<11.5$ g/dL for women aged ≥19 years [21]) and tuberculosis. Tuberculosis was defined as the detection of acid-fast bacilli or positive Xpert MTB/RIF assay (Cepheid, Sunnyvale, CA, USA) from sputum or an extra-pulmonary sample, or prescription of anti-tuberculosis medication in the presence of an International Classification of Diseases (ICD)-10 code indicating tuberculosis (A15-A19) or clinical signs suggestive of tuberculosis and the start of anti-tuberculosis drugs within 3 months before and after ART initiation. Tuberculosis was considered unlikely if no prescription of anti-tuberculosis drugs and no clinical diagnosis was provided. In all other cases, tuberculosis was indeterminate and treated as missing data in the analyses. Baseline WHO stage and CD4 cell counts were those closest to ART initiation.

## Accessibility to identifiable data

The dataset is only accessed by the principal investigator and the statistician (leading author). The dataset used for this study is de-identified using unique patient numbers to ensure the privacy of participants. Only the senior authors have access to identifiable data the study.

## Statistical analysis

Summary statistics were used to describe patients' characteristics at ART initiation. Systematic differences between patients with and without follow-up BMI measurements were assessed

using Chi-Square tests or Fisher's exact tests as appropriate. Subsequent analyses were complete case, based on the BMI measurements observed. We plotted BMI evolution over time for patients with at least one follow-up measurement by baseline BMI category. Average trends were estimated using cubic splines with 60 knots [22]. The distribution of BMI categories over time was illustrated using stacked graphs truncated at 48 months. Average BMI was used for patients with multiple BMI measurements in a given calendar month.

BMI was analyzed as a continuous outcome using a stratified piecewise linear mixed-effects model [23–25]. The model was stratified by three major baseline BMI categories (that is underweight, normal BMI, and overweight/obese) with separate slopes for the time before and after 9 months from ART initiation. The 9 months' time point was based on a visual assessment of the BMI trends. In sensitivity analyses, we considered alternative time points of 6, 7, 8, and 12 months. The likelihood ratio test was used to assess the need for both random intercept and slope variance components [23]. The final model included both random intercepts and slopes with robust standard errors, unstructured random effect, and simple residual (regression residuals) variance-covariance structures. In modeling, BMI was transformed using a natural logarithm, reflecting the percentage change of actual BMI measurements.

Extended cumulative Kaplan-Meier estimates with time-updated BMI categories were used to illustrate time to death/LTFU by baseline BMI category [26, 27]. Cox proportional hazard models were used to investigate the association between time to death/LTFU and BMI category as a time-updated covariate, adjusted for baseline covariates [28, 29]. Participants with missing data on baseline covariates were excluded from the respective models. Time-updated BMI was the value recorded at the most recent visit. Follow-up in months was measured from ART initiation to death, LTFU, or transfer to another clinic. Those who remained in active care were censored on the date of database closure (30th June 2019).

Data analyses were done in SAS version 9.4 (SAS Institute Inc., Cary, North Carolina, USA) and Stata version 14 (StataCorp LP, College Station, Texas, USA).

### Ethical considerations

The KIULARCO research cohort obtained ethical approval from the Ifakara Health Institute Review Board (IHI/IRB/No:16–2006) and the National Health Research Committee of the National Institute for Medical Research of Tanzania (NIMR/HQ/R.8a/Vol. IX/620). At enrolment, all patients are informed and asked to participate in KIULARCO. A written informed consent is obtained. All signed consent forms are stored in patients' files at the clinic.

## Results

### Patients' characteristics at baseline

From January 2013 to December 2018, 4,381 patients were enrolled in KIULARCO. Of these, 2,252 patients were excluded for the following reasons: aged <19 years, transit patients, prior exposure to ART, not on ART during the study period, missing baseline BMI, and pregnancy (Fig 1).

Of 2,129 participants at baseline, 398 (19%) patients were underweight, and 356 (17%) were overweight/obese (Table 1). Within the extremes of malnutrition, 183 (9%) patients were moderately or severely underweight, and 101 (5%) were obese. The majority of patients were female (n = 1249; 59%), aged 35–44 years (n = 779; 37%), farmers (n = 1827; 86%), and had a primary school education (n = 1773; 83%). Approximately half of the patients (n = 1025; 48%) had a WHO stage 3/4 and 533 (25%) had a CD4 cell count <100 cells/mm$^3$.

The distribution of sex, age, occupation, education, CD4 cell count, tuberculosis status, and hypertension status was similar for patients with (n = 1975; 93%) or without (n = 154; 7%) a

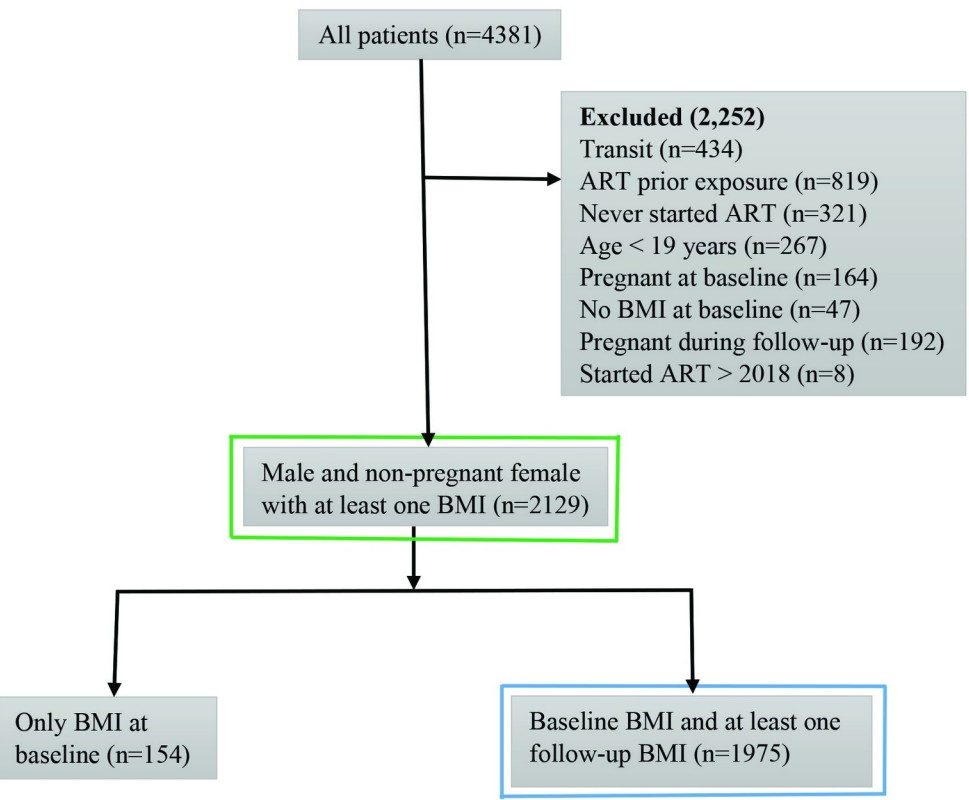

**Fig 1. Inclusion of KIULARCO participants enrolled from January 2013 to December 2018, with follow-up through end of June 2019.** The baseline and death or LTFU analyses used patients with at least one BMI measurement (n = 2129 patients, as indicated in green) and the trend analyses used patients with baseline BMI and at least one follow-up BMI (n = 1975 patients, as indicated in blue).

follow-up BMI measurement (S1 Table). Participants without follow-up BMI measurements were more likely to have been initiated on a protease inhibitor-based ART regimen, and to have more advanced WHO stage 3/4 and anemia, compared to those with follow-up BMI measurements.

## BMI trends

The median follow-up time from ART initiation to the last BMI measurement was 20 months (interquartile range (IQR): 6–42; maximum 79). Overall, 22,027 BMI measurements were assessed. Each patient contributed a median of 9 measurements (IQR: 5–15; range 1–48). In the 1,975 patients with at least two measurements, BMI increased on average during the first 6–12 months after ART initiation, and the values stabilized after 12 months (Fig 2a). The initial increase was most pronounced among patients who were underweight at baseline (Fig 2b). Most patients with a normal and overweight/obese BMI at baseline remained in the same BMI category during follow-up (Fig 2c and 2d).

Among patients starting ART while underweight, the majority (80%) reached a normal BMI within 6–8 months following ART initiation (Fig 3b). Very few patients (5–10%) moved to the overweight/obese BMI category. Among the 108 (30%) patients remaining underweight throughout the study period, the follow-up time was short (median 2 months, IQR: 1–8).

**Table 1. Patients' characteristics at ART initiation by BMI categories.**

| Characteristics | Normal BMI | Underweight | Overweight or Obese | Total |
|---|---|---|---|---|
| Overall | 1375 (64.6) | 398 (18.7) | 356 (16.7) | 2129 (100) |
| Sex | | | | |
| Male | 631 (45.9) | 168 (42.2) | 81 (22.8) | 880 (41.3) |
| Female | 744 (54.1) | 230 (57.8) | 275 (77.3) | 1249 (58.7) |
| Age in years | | | | |
| 19–34 | 419 (30.5) | 114 (28.6) | 98 (27.5) | 631 (29.6) |
| 35–44 | 495 (36.0) | 142 (35.7) | 142 (39.9) | 779 (36.6) |
| 45 and above | 461 (33.5) | 142 (35.7) | 116 (32.6) | 719 (33.8) |
| Occupation | | | | |
| Non-farmers | 198 (14.4) | 40 (10.1) | 64 (18.0) | 302 (14.2) |
| Farmers | 1177 (85.6) | 358 (90.0) | 292 (82.0) | 1827 (85.8) |
| Education | | | | |
| No education | 151 (11.0) | 47 (11.8) | 31 (8.7) | 229 (10.8) |
| Primary school | 1140 (82.9) | 336 (84.4) | 297 (83.4) | 1773 (83.3) |
| Above primary school | 84 (6.1) | 15 (3.8) | 28 (7.9) | 127 (6.0) |
| ART | | | | |
| First line[a] | 1360 (98.9) | 396 (99.5) | 354 (99.4) | 2110 (99.1) |
| Second line[b] | 15 (1.1) | 2 (0.5) | 2 (0.6) | 19 (0.9) |
| CD4 count, cells/μL | | | | |
| Below 100 | 317 (23.1) | 170 (42.7) | 46 (12.9) | 533 (25.0) |
| 100–199 | 263 (19.3) | 73 (18.3) | 54 (15.2) | 390 (18.3) |
| 200–349 | 349 (25.4) | 58 (14.6) | 96 (27.0) | 503 (23.6) |
| 350 and above | 307 (22.3) | 43 (10.8) | 130 (36.5) | 480 (22.6) |
| Missing data | 139 (10.1) | 54 (13.6) | 30 (8.4) | 223 (10.5) |
| WHO stage | | | | |
| Stage 1/2 | 722 (52.5) | 86 (21.6) | 263 (73.9) | 1071 (50.3) |
| Stage 3/4 | 629 (45.8) | 310 (77.9) | 86 (24.2) | 1025 (48.1) |
| Missing data | 24 (1.8) | 2 (0.5) | 7 (2.0) | 33 (1.6) |
| Tuberculosis | | | | |
| Negative | 1088 (79.1) | 267 (67.1) | 320 (89.9) | 1675 (78.7) |
| Positive | 241 (17.5) | 119 (29.9) | 30 (8.4) | 390 (18.3) |
| Missing data | 46 (3.4) | 12 (3.0) | 6 (1.7) | 64 (3.0) |
| Arterial hypertension | | | | |
| No arterial hypertension | 1232 (89.6) | 371 (93.2) | 294 (82.6) | 1897 (89.1) |
| Arterial hypertension | 109 (7.9) | 11 (2.8) | 57 (16.0) | 177 (8.3) |
| Missing data | 34 (2.5) | 16 (4.0) | 5 (1.4) | 55 (2.6) |
| Anemia status | | | | |
| Not anemic | 415 (30.2) | 56 (14.1) | 186 (52.3) | 657 (30.9) |
| Anemic | 808 (58.8) | 299 (75.1) | 130 (36.5) | 1237 (58.1) |
| Missing data | 152 (11.1) | 43 (10.8) | 40 (11.2) | 235 (11.0) |

Column percentages are presented. percentages may be slightly below or above 100 due to rounding.

[a]First-line ART was AZT+3TC+NVP, AZT+3TC+EFV, TDF+FTC+EFV, TDF+FTC+NVP, TDF+3TC+EFV, ABC+3TC+EFV.

[b]Second-line ART was TDF+FTC+LPV/r, AZT+3TC+LPV/r, TDF+FTC+ATV/r.

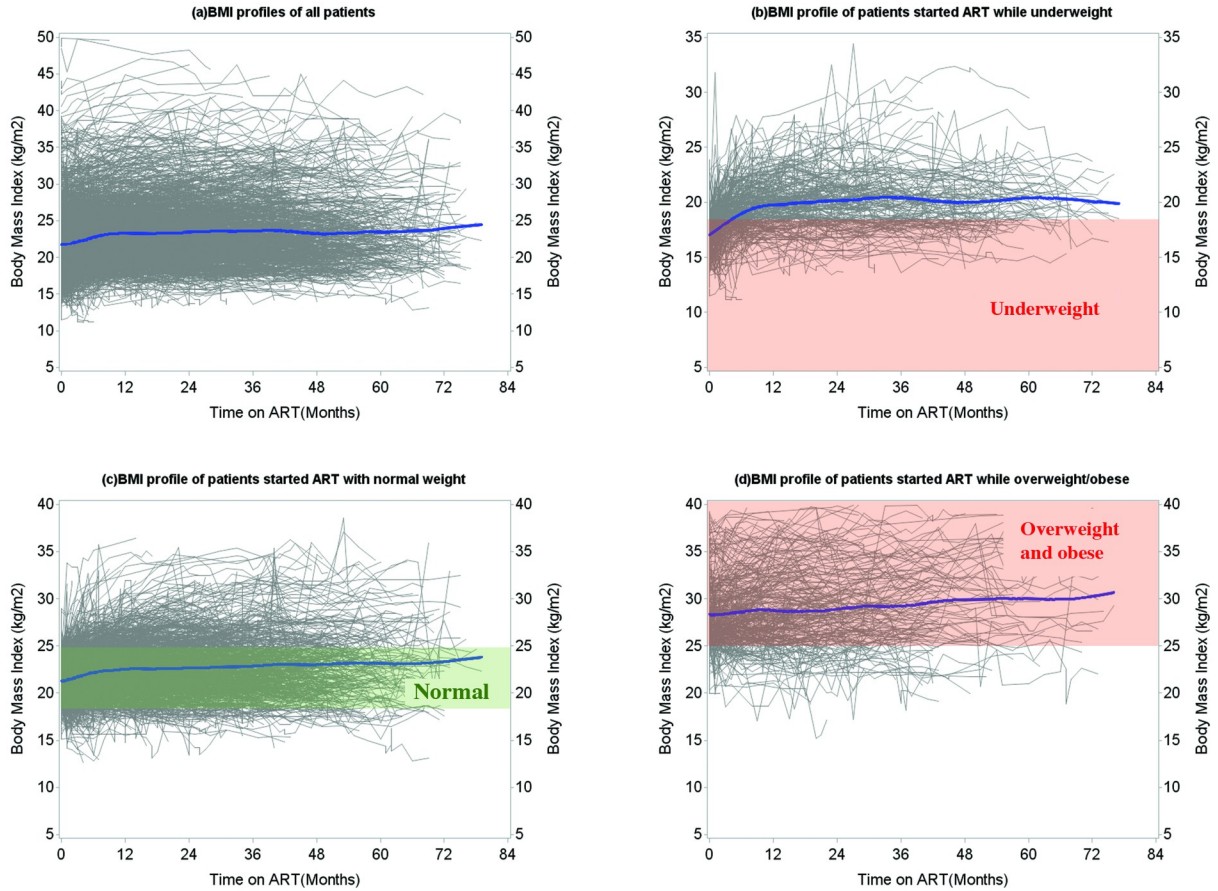

**Fig 2. BMI trends over time following ART initiation (based on continuous BMI).** Graphs include BMI profiles of all patients with at least one follow-up BMI measurement (n = 1,975). (a) Overall BMI profiles for all patients, and (b-d) BMI profiles stratified by baseline BMI categories. Each line represents an individual patient (grey-coloured). The bold lines represent the average BMI trend estimated using splines (blue-coloured).

Compared to patients who did not remain underweight throughout, often, patients who remained underweight died (n = 24; 22% versus n = 6; 2%) or became LTFU (n = 51; 47% versus n = 50; 20%). More than half of the participants who had either a normal BMI (n = 667; 52%) or were overweight/obese (n = 228; 68%) at baseline remained in the same BMI category throughout follow-up (Fig 3c and 3d).

## Association between baseline characteristics and BMI during follow-up

Across the three baselines BMI categories, there was an increase in BMI during the first 9 months following ART initiation (Table 2). During this period, for every three additional follow-up months, BMI increased by 0.7% among patients who had normal baseline BMI (effect = 1.007; 95% confidence interval (CI) 1.005–1.008; P<0.0001), 2% among patients who were underweight at baseline (effect = 1.02; 95% CI 1.01–1.02; P<0.0001), and 0.4% among patients who were overweight/obese at baseline (effect = 1.004; 95% CI 1.002–1.006; P = 0.001). From 9 months onwards, for every three additional months, there was a decrease in BMI of 0.2% among patients having normal BMI at baseline (effect = 0.998; 95% CI 0.996–1.00; P = 0.05) and of 0.4% among underweight patients (effect = 0.996; 95% CI 0.993–0.999;

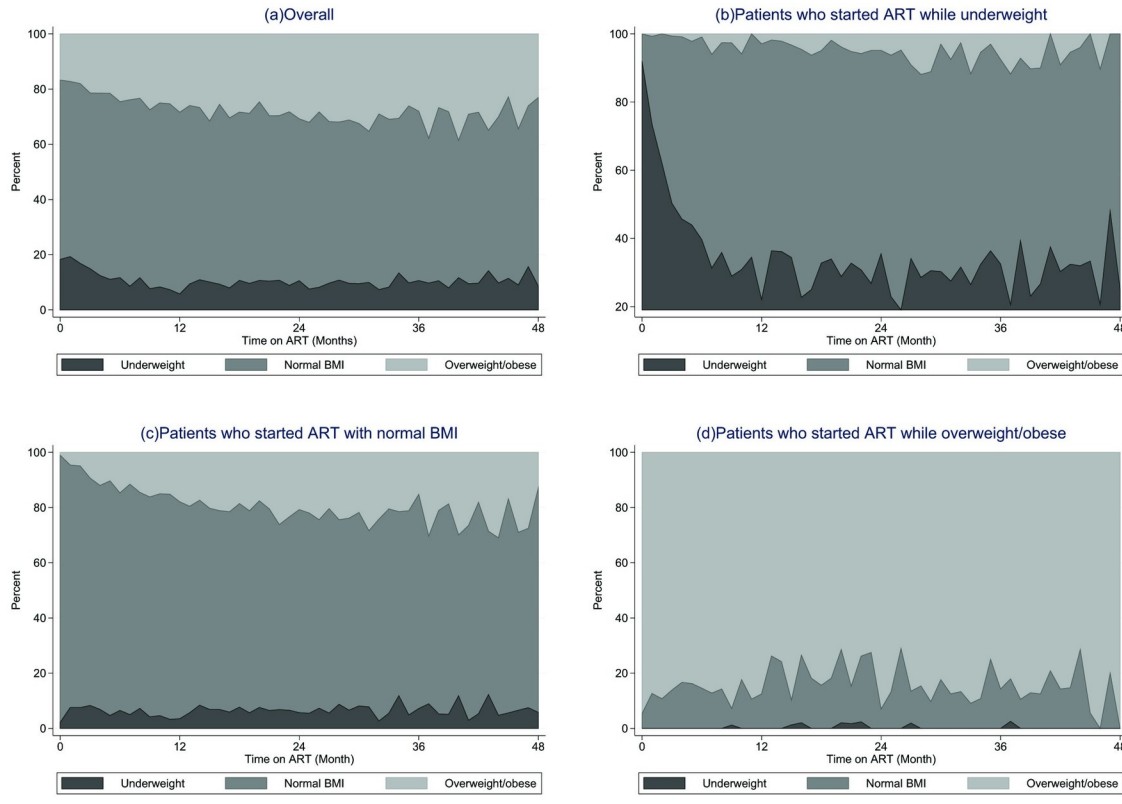

**Fig 3. BMI trends over time following ART initiation (based on categorical BMI).** The trend of BMI categories over time among patients with at least one follow-up BMI (n = 1975). Panel (a) Overall plot regardless of baseline BMI category, panel (b-d) BMI categories proportions by baseline BMI categories.

P = 0.003). There was no evidence of a change in BMI from 9 months onwards among patients who were overweight/obese at baseline (effect = 1.00; 95% CI: 0.996–1.003; P = 0.84). Changes in BMI slopes in participants who were overweight/obese at baseline were similar across different time points (S2 Table).

Among patients with normal BMI or overweight/obese at baseline, being female was associated with a higher follow-up BMI compared to male patients, while there was no evidence of a difference in follow-up BMI by sex among those underweight at baseline (Table 2). There were no clear associations between follow-up BMI and age, except that among patients who were overweight/obese at baseline, those aged 19–34 years had lower follow-up BMI compared to older patients. For farmers, there was a trend of lower follow-up BMI compared to non-farmers across all three baseline BMI categories. In patients having a normal BMI at baseline, a WHO stage 1/2 was associated with a higher follow-up BMI compared to participants with a WHO stage 3/4. Among patients overweight/obese at baseline, tuberculosis was associated with a lower follow-up BMI compared to patients without tuberculosis. There was a trend towards lower follow-up BMI among anemic compared to non-anemic patients across all three BMI categories.

## Association between BMI and death or loss of follow-up

Over the course of follow-up, 107 (5%) participants died, 592 (28%) were LTFU, 228 (11%) were transferred to another clinic, and 1202 (57%) remained in HIV care (S3 Table). The

**Table 2. Association between patients' characteristics at ART initiation and follow-up BMI, stratified by BMI at ART initiation.**

| Characteristics | Normal BMI (n = 1020) | | Underweight (n = 287) | | Overweight or obese (n = 275) | |
|---|---|---|---|---|---|---|
| | Estimate (95% Confidence interval) | P-value | Estimate (95% Confidence interval) | P-value | Estimate (95% Confidence interval) | P-value |
| Time effect (per 3 months) | | | | | | |
| Before up to 9 months | 1.007 [1.005, 1.008] | < .0001 | 1.016 [1.013, 1.018] | < .0001 | 1.004 [1.002, 1.006] | 0.001 |
| After 9 months | 0.998 [0.996, 1.000] | 0.05 | 0.996 [0.993, 0.999] | 0.003 | 1.000 [0.996, 1.003] | 0.842 |
| Sex | | | | | | |
| Male | Ref | | Ref | | Ref | |
| Female | 1.01 [1.003, 1.03] | 0.02 | 0.98 [0.95, 1.01] | 0.16 | 1.08 [1.04, 1.11] | < .0001 |
| Age (years) | | | | | | |
| 19–34 | 1.00 [0.99, 1.02] | 0.81 | 1.01 [0.98, 1.05] | 0.52 | 0.95 [0.91, 0.998] | 0.04 |
| 35–44 | 1.01 [1.00, 1.03] | 0.08 | 0.99 [0.96, 1.02] | 0.56 | 1.01 [0.97, 1.06] | 0.52 |
| ≥ 45 | Ref | | Ref | | Ref | |
| Occupation | | | | | | |
| Non-farmers | Ref | | Ref | | Ref | |
| Farmers | 0.97 [0.95, 0.99] | 0.003 | 0.96 [0.91, 1.02] | 0.17 | 0.95 [0.90, 1.00] | 0.06 |
| Education | | | | | | |
| No education | Ref | | Ref | | Ref | |
| Primary school | 1.00 [0.98, 1.03] | 0.64 | 0.98 [0.93, 1.03] | 0.40 | 1.03 [0.97, 1.09] | 0.37 |
| Above primary school | 1.01 [0.97, 1.04] | 0.73 | 0.91 [0.83, 1.00] | 0.06 | 1.06 [0.96, 1.18] | 0.27 |
| CD4 count, cells/μL | | | | | | |
| < 100 | Ref | | Ref | | Ref | |
| 100–199 | 1.01 [0.99, 1.03] | 0.44 | 1.02 [0.99, 1.06] | 0.18 | 1.00 [0.94, 1.06] | 0.92 |
| 200–349 | 1.01 [1.00, 1.03] | 0.13 | 1.01 [0.96, 1.05] | 0.76 | 1.02 [0.97, 1.07] | 0.42 |
| ≥ 350 | 1.00 [0.98, 1.02] | 0.81 | 0.99 [0.95, 1.03] | 0.76 | 1.01 [0.96, 1.06] | 0.69 |
| WHO stage | | | | | | |
| Stage 1/2 | 1.02 [1.002, 1.03] | 0.02 | 1.01 [0.98, 1.04] | 0.60 | 0.99 [0.95, 1.04] | 0.77 |
| Stage 3/4 | Ref | | Ref | | Ref | |
| Tuberculosis | | | | | | |
| Negative | Ref | | Ref | | Ref | |
| Positive | 1.00 [0.98, 1.02] | 0.81 | 1.00 [0.97, 1.04] | 0.82 | 0.94 [0.88, 0.996] | 0.04 |
| Arterial hypertension | | | | | | |
| No arterial hypertension | Ref | | Ref | | Ref | |
| Arterial hypertension | 1.01 [0.99, 1.03] | 0.35 | 1.02 [0.93, 1.12] | 0.66 | 1.00 [0.95, 1.04] | 0.83 |
| Anemia status | | | | | | |
| Not anemic | Ref | | Ref | | Ref | |
| Anemic | 0.99 [0.97, 1.00] | 0.07 | 0.96 [0.93, 0.99] | 0.02 | 0.97 [0.94, 1.00] | 0.08 |

Estimates based on an adjusted stratified linear mixed model and time modelled using piecewise regression, among patients with non-missing data at ART initiation. Empirical standard error was used.

The BMI outcome was transformed using a natural logarithm, and the estimates should be interpreted in terms of the percent change of actual BMI.

"Ref" indicates the reference category of the modelled explanatory variable.

median time to death was 3 months (IQR 1–10), 8 months (IQR 5–21) for LTFU, and 7 months (IQR 1–23) for transferred out. For those remaining in care, the median follow-up time was 38 months (IQR 21–53). Compared to patients retained in care, patients who died during follow-up tended to be aged ≥45 years (n = 44; 41% versus n = 412; 34%), were more

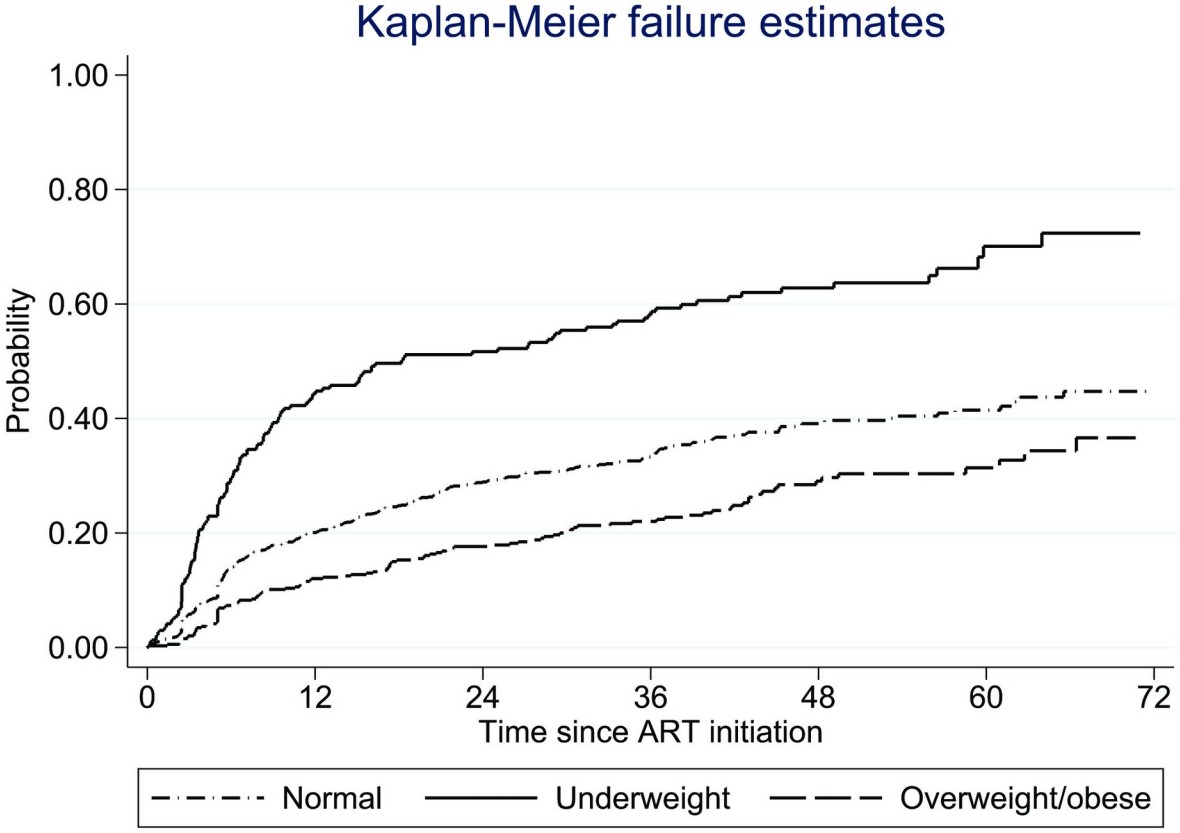

**Fig 4. The cumulative survival function for death/LTFU events by time-updated BMI.**

frequently underweight (n = 36; 34% versus n = 184; 15%), and were more likely to have low CD4 cell count <100 cells/µL (n = 56; 52% versus n = 262; 22%), WHO stage 3/4 (n = 81; 76% versus n = 485; 40%), and co-morbidities (i.e., tuberculosis (n = 37; 35% versus n = 191; 16%) and anaemia (n = 85; 79% versus n = 634; 53%)). Compared to patients retained in care, those who were LTFU were also more likely to have a WHO stage 3/4 and anaemic.

Underweight patients had a higher probability of death/LTFU compared to those with normal BMI or those being overweight/obese (Fig 4). The cumulative probabilities of death/LTFU by five years were 0.41 (95% CI 0.38–0.45) for participants with normal BMI, 0.70 (95% CI 0.62–0.78) for those underweight, and 0.31 (95% CI 0.26–0.37) for those overweight/obese.

Adjusting for confounders and compared to normal BMI at the most recent visit, being underweight was associated with a higher risk of death/LTFU (hazard ratio (HR) = 2.04; 95% CI 1.64–2.53), while being overweight/obese was associated with a lower risk (HR = 0.72; 95% CI 0.56–0.93) (Table 3). The risk of death/LTFU was higher among those with no education, WHO stage 3/4, and anemic.

## Discussion

In this study assessing longitudinal BMI trends in ART-naïve PLHIV starting ART in rural Africa, we found a double burden of malnutrition with 398 (19%) underweight and 356 (17%) overweight/obesity at ART initiation. BMI increased in the first 9 months on ART in all patients, but with larger increases in those who were underweight at baseline. Predictors of a

**Table 3. Association between death or loss to follow-up and patient characteristics at ART initiation and time-updated BMI categories (n = 1690).**

| Characteristics | Adjusted Hazard Ratio [95% Confidence Interval] | P-value |
|---|:---:|:---:|
| Time updated BMI | | |
| Underweight | 2.04 [1.64, 2.53] | <0.0001 |
| Normal | Ref | |
| Overweight and obese | 0.72 [0.56, 0.93] | 0.01 |
| Sex | | |
| Male | Ref | |
| Female | 0.86 [0.71, 1.03] | 0.10 |
| Age in years | | |
| 19–34 | 1.20 [0.97, 1.50] | 0.10 |
| 35–44 | 0.92 [0.74, 1.14] | 0.44 |
| ≥ 45 | Ref | |
| Occupation | | |
| Non-farmers | Ref | |
| Farmers | 0.96 [0.72, 1.28] | 0.79 |
| Education | | |
| No education | Ref | |
| Primary school | 0.72 [0.55, 0.95] | 0.02 |
| Above primary school | 0.77 [0.48, 1.25] | 0.30 |
| CD4 count, cells/µL | | |
| < 100 | Ref | |
| 100–199 | 0.78 [0.61, 1.01] | 0.06 |
| 200–349 | 0.81 [0.63, 1.03] | 0.08 |
| ≥ 350 | 1.03 [0.80, 1.33] | 0.81 |
| WHO stage | | |
| Stage 1/2 | 0.65 [0.53, 0.80] | <0.0001 |
| Stage 3/4 | Ref | |
| Tuberculosis | | |
| Negative | Ref | |
| Positive | 0.90 [0.71, 1.13] | 0.36 |
| Hypertension | | |
| No arterial hypertension | Ref | |
| Arterial hypertension | 0.90 [0.64, 1.26] | 0.54 |
| Anemia status | | |
| No anemia | Ref | |
| Anemic | 1.31 [1.06, 1.62] | 0.01 |

Based on 1690 patients with non-missing data at ART initiation.

All characteristics are measured at ART initiation, while BMI is time-updated according to the value at the previous visit.

Cox proportional hazard model was used for multivariable analysis

"Ref" indicates the reference category of the modeled variable.

higher BMI during follow-up were female sex and less advanced WHO stage, while being a farmer and being anemic were associated with a lower follow-up BMI. Mortality was low (n = 107; 5%), but the loss to follow-up was high (n = 592; 28%), in line with previous papers [30, 31]. Being underweight, having no education, having WHO stage 3/4, and having anemia were associated with an increased risk of death/LTFU.

The high percentage of underweight patients at ART start (19%) has been shown in other settings in sub-Saharan Africa. One study from South Africa showed a prevalence of 13% in mixed urban, semi-urban, and rural areas [32]. A recent meta-analysis from sub-Saharan Africa showed a pooled prevalence of undernutrition of 24% with studies originating mostly from Ethiopia [33]. Partly, this might be explained by the fact that 49% of patients had a WHO stage 3/4 and 49% had a CD4 cell count <200 cells/mm$^3$, both factors associated with under-nutrition [33]. Untreated HIV infection exerts a high metabolic demand and is associated with low BMI, and in the worst case, wasting syndrome [6, 7]. Opportunistic infections and malig-nancies may further decrease BMI [7]. On the other hand, being underweight in our setting might also reflect high levels of background poverty, food insecurity, and additional non-HIV-related infections [5, 34].

While the prevalence of underweight remains common in low- and middle-income coun-tries, the prevalence of overweight and obesity is increasing among PLHIV. In our study, over-weight/obesity was at 17% which is lower compared to almost 30% in a study done in the urban setting of Mwanza, Tanzania [35]. Studies conducted in South Africa and Ivory Coast and South Africa also showed a higher prevalence of overweight (26%, 20% and 37%, respec-tively) [32, 36, 37]. These findings are similar to pooled estimates of a meta-analysis conducted among PLHIV living in low- and middle-income countries [38]. Reasons for the lower preva-lence of overweight/obesity in our setting might be the rural character with associated higher levels of poverty.

The increase in BMI during the first months on ART has been documented in a similar study conducted in South Africa [39], with 50% of patients having an increase in BMI during the first 12 months on ART. This clinically important change in BMI for a majority of patients —those starting with a low BMI—indicates a return to health effects and it is due to the immune reconstitution preventing opportunistic infections and wasting [8, 37]. On the other hand, BMI increases to overweight/obesity observed in patients with a normal weight at base-line are a risk factor for non-communicable diseases such as cardiovascular diseases and diabe-tes [1, 2]. PLHIV–due to a combination of HIV-induced chronic immune activation, endothelial dysfunction, and possible metabolic side-effects of ART–are already at increased risk of cardiovascular disease [6, 15, 40–42]. Interestingly, in our cohort, BMI stabilized after an initial increase, which was different from findings in a cohort from Johannesburg, South Africa, where BMI gradually increased for over 8 years [39]. One possible explanation for the differences could be due to the settings: in contrast to the urban setting of Johannesburg, our study setting is rural and most of our patients are farmers with a physically active lifestyle.

The mortality rate of 5% in this cohort is relatively low, while a substantial proportion (23%) were LTFU. In a previous study conducted in the KIULARCO cohort, 40% of patients who were LTFU and traced were found to have died [31]. As expected, patients who were underweight at ART initiation had an increased risk of reaching the combined endpoint of death/LTFU, which is in line with a study conducted in Lusaka, Zambia [8]. In contrast, over-weight/obese patients had a lower hazard of death or LTFU compared to patients with normal BMI. These findings were similar to a study from South Africa in participants living in an urban setting [36]. More studies with longer follow-up time are needed to investigate a possi-ble association between being overweight and cardiovascular endpoints and mortality in sub-Saharan Africa.

A strength of this study is the systematic, prospective collection of longitudinal routine clin-ical data, including BMI, in a large rural African cohort of PLHIV before the rollout of inte-grase-inhibitors, creating a baseline for weight changes on non-nucleoside reverse transcriptase inhibitor-based-based treatments. The study has several limitations. Information on dietary patterns and hip/waist circumference was not captured routinely, which could

better describe cardiometabolic risk [43]. BMI trend analyses could be done only in those with at least two BMI measurements, possibly introducing selection bias, as those with only one BMI measurement tended to have more advanced HIV disease and were more likely to have died. Further, we included only patients that survived to start ART, and thus our findings are not necessarily generalizable to all PLHIV. The assessment of morbidity and mortality associated with overweight/obesity requires long follow-up periods, therefore our relatively short follow-up time might not be enough to accurately capture these associations. Lastly, we did not analyze the impact of different ART regimens on BMI, as almost all patients were on an efavirenz-based first-line regimen. Dolutegravir, which has been reported to be associated with weight gain [15, 44–48], has only relatively recently been rolled out in Tanzania. Future studies are planned to assess the impact of dolutegravir on BMIs in this population and the current study will set a baseline to be compared with the dolutegravir-based ART regimen.

## Conclusion

We found a double burden of malnutrition in this rural HIV cohort of Tanzania with important implications for clinical outcomes. Interventions to address various states of malnutrition and their underlying causes should be considered in HIV clinics. While our data suggest that higher BMI is associated with improved survival, more research is needed, particularly in the context of earlier initiation of ART since 2018 and newer antiretrovirals like dolutegravir being introduced.

## Supporting information

**S1 Table. Patients' characteristics at ART initiation by the availability of follow-up BMI measurements (numbers with their respective percentages).**
(PDF)

**S2 Table. Assessment of different change points.**
(PDF)

**S3 Table. Patient characteristics at ART initiation by outcomes (numbers with their respective percentages).**
(PDF)

## Acknowledgments

We thank the staff of the Chronic Disease Clinic of the St Francis Referral Hospital, Ifakara, Tanzania. We are grateful to all the participants of the Kilombero and Ulanga Antiretroviral Cohort (KIULARCO).

### The Kilombero and Ulanga Antiretroviral Cohort study group (KIULARCO)

Aschola Asantiel[1], Farida Bani[1], Manuel Battegay[2], Theonestina Byakuzana[1], Joyce Claud[1], Adolphina Chale[1], Elizabeth Dotto[1], Gideon Francis[3], Tracy R. Glass[4,5], Yvonne Haridas[1,3], Speciosa Hwaya[3], Aneth V Kalinjuma[1,6,7], Andrew Katende[1], Amiri Kayera[2], Yassin Kisunga[1], Olivia Kitau[1], Bernard Kivuma[1], Thomas Klimkait[6], Juma Kupewa[1], Namsifueli J Ley[1], Ezekiel Luoga[1], Jerome Lwali[1], Swalehe Masoud[1], Mohammed Mbaruku[1], Geofrey Mbunda[1], Josephine Mhina[1], Slyakus Mlembe[1], Mengi Mkulila[2], Margareth Mkusa[2], Lina Mnunga[2], Alpha Mninje[2], Dorcas K Mnzava[1], Getrud J Mollel[1], Lilian Moshi[1], Germana Mossad[2], Dolores Mpundunga[2], Athumani Mtandanguo[1], Elizabeth Mwambashi[2], Selerine Myeya[1], Sanula

Nahota[1], Sharifa Nakapala[2], Regina Ndaki[1], Robert C. Ndege[1], Suzan Ngahyoma[3], Agatha Ngulukila[1], Alex John Ntamatungiro[1,6,7], Amina Nyuri[1], James Okuma[4,5], Daniel H Paris[4,5], Martin Rohacek[1,4,5], Petro Togolani Sabuni[1], Leila Samson[1], Elizabeth Senkoro[1], George Sigalla[1], Jamali B Siru[1], Jenifa Tarimo[1], Juerg Utzinger[4,5], Fiona Vanobberghen[4,5], Maja Weisser[1,4,5,8], John Wigay[1], Herieth Wilson[1], Lulu Wilson[1]

 1 Interventions and Clinical Trials Department, Ifakara Health Institute, Ifakara, Tanzania
 2 University Hospital Basel, Basel, Switzerland
 3 Saint Francis Referral Hospital, Ifakara, Tanzania
 4 Swiss Tropical and Public Health Institute, Allschwil, Switzerland
 5 University of Basel, Basel, Switzerland
 6 Epidemiology and Biostatistics Department, School of Public Health, Faculty of Health Sciences, University of the Witwatersrand
 7 Division of Public Health, School of Public Health and Family Medicine, University of Cape Town, South Africa
 8 Division of Infectious Diseases and Hospital Epidemiology, University Hospital Basel, University of Basel, Switzerland
 Maja Weisser is the Research cordinator of the KIULARCO study group
 Email: mweisser@ihi.or.tz

## Author Contributions

**Conceptualization:** Aneth Vedastus Kalinjuma, Hannah Hussey, Getrud Joseph Mollel, Emilio Letang, Manuel Battegay, Tracy R. Glass, Daniel Paris, Fiona Vanobberghen, Maja Weisser.

**Data curation:** Aneth Vedastus Kalinjuma, Tracy R. Glass, Fiona Vanobberghen.

**Formal analysis:** Aneth Vedastus Kalinjuma.

**Methodology:** Aneth Vedastus Kalinjuma, Fiona Vanobberghen.

**Software:** Aneth Vedastus Kalinjuma, Fiona Vanobberghen.

**Supervision:** Manuel Battegay, Fiona Vanobberghen, Maja Weisser.

**Writing – original draft:** Aneth Vedastus Kalinjuma, Hannah Hussey.

**Writing – review & editing:** Aneth Vedastus Kalinjuma, Hannah Hussey, Getrud Joseph Mollel, Emilio Letang, Manuel Battegay, Tracy R. Glass, Daniel Paris, Fiona Vanobberghen, Maja Weisser.

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
