## [Decision Letter · Decision Letter 0]

18 May 2023

PONE-D-23-06462Body mass index trends and its impact of under and overweight on outcome among PLHIV on antiretroviral treatment in rural Tanzania: A prospective cohort studyPLOS ONE

Dear Dr. Kalinjuma,

Thank you for submitting your manuscript to PLOS ONE. After careful consideration, we feel that it has merit but does not fully meet PLOS ONE’s publication criteria as it currently stands. Therefore, we invite you to submit a revised version of the manuscript that addresses the points raised during the review process.

We look forward to receiving your revised manuscript.

Kind regards,

I. Marion Sumari-de Boer, Ph.D

Academic Editor

PLOS ONE

“I have read the journal’s policy and the author of this manuscript has the following competing interests: Emilio Letang is a full-time employee ViiV Healthcare since May 2021. Other authors have declared that no competing interests exist.”

3. One of the noted authors is a group or consortium [KIULARCO study group]. In addition to naming the author group, please list the individual authors and affiliations within this group in the acknowledgments section of your manuscript. Please also indicate clearly a lead author for this group along with a contact email address.

Reviewers' comments:

Reviewer's Responses to Questions

**Comments to the Author**

1. Is the manuscript technically sound, and do the data support the conclusions?

Reviewer #1: Yes

Reviewer #2: Yes

2. Has the statistical analysis been performed appropriately and rigorously? 

Reviewer #1: Yes

Reviewer #2: Yes

3. Have the authors made all data underlying the findings in their manuscript fully available?

Reviewer #1: Yes

Reviewer #2: Yes

4. Is the manuscript presented in an intelligible fashion and written in standard English?

Reviewer #1: Yes

Reviewer #2: Yes

5. Review Comments to the Author

Reviewer #1: The manuscript presents a very important topic and is well written, however there are few things which needs to be improved or clarified

1. In the abstract section : BMI categories need to clearly stated eg: the author just mentioned overweight / obese of 25.0kg/m2. this needs to be clear if it is greater than 25 or just 25 kg/m2

2. Double burden of malnutrition should be clearly defined, and the author should be consistent throughout if it is double burden of malnutrition of just underweight and overweight

3. Under Table 1, the author should clarify why she/ he included the information that column percentages are presented of non missing data (this is confusing)

4. Lines 247 -249 the word "their category" need to be specified which category the author is referring

5. Line 254 - thin line and thick lines for figure 1 needs to be more clear, as it is difficult to differentiate which line is thin and which one is thick

6. Lines 284 - 285 which results the author is reefing needs be clear

7. Lines 293 - 294 WHO stages needs to be defined clearly

8. Lines 309 It might be important for the author to mention which comorbidities he / she is referring

9. Lines 362 - 363 The sentence should be competed, From another study.....which study ??

Reviewer #2: Thank you very much for the opportunity to read this manuscript. I enjoyed reading it as the overall quality of the study and the writing is excellent. All of my comments following below are suggestions that the authors may want to consider in a revised manuscript.

Introduction

Although this is an understudied area, there has been much research on this topic in low-to-middle-income countries over the past couple of years that has not been included in the introduction, although this is indeed largely in urban areas.

Alebel, A., Demant, D., Petrucka, P., & Sibbritt, D. (2021). Effects of undernutrition on mortality and morbidity among adults living with HIV in sub-Saharan Africa: a systematic review and meta-analysis. BMC infectious diseases, 21, 1-20.

Fuseini, H., Gyan, B. A., Kyei, G. B., Heimburger, D. C., & Koethe, J. R. (2021). Undernutrition and HIV infection in sub-Saharan Africa: health outcomes and therapeutic interventions. Current HIV/AIDS Reports, 18, 87-97.

Seid, A., Seid, O., Workineh, Y., Dessie, G., & Bitew, Z. W. (2023). Prevalence of undernutrition and associated factors among adults taking antiretroviral therapy in sub-Saharan Africa: A systematic review and meta-analysis. Plos one, 18(3), e0283502.

Mahlangu, K., Modjadji, P., & Madiba, S. (2020, August). The nutritional status of adult antiretroviral therapy recipients with a recent HIV diagnosis; a cross-sectional study in primary health facilities in Gauteng, South Africa. In Healthcare (Vol. 8, No. 3, p. 290). MDPI.

Methods

It would be great to know more about the districts where this data has been collected, particularly considering the limitations of the current body of evidence discussed in the introduction.

The authors may also want to comment on the quality of data collection within healthcare services in Tanzania.

The statistical analyses have been described well and are appropriate for the outcomes.

Results

The results are overall well-written. Was an analysis undertaken between those remaining in care and those who where LTFU? Are there differences in demographic characteristics?

Discussion

The discussion is particularly well developed but would benefit from further comparisons with similar studies in other African settings.

6. PLOS authors have the option to publish the peer review history of their article (what does this mean?). If published, this will include your full peer review and any attached files.

Reviewer #1: **Yes: **Mary Vincent Mosha

Reviewer #2: No

---

## [Author Response · Author response to Decision Letter 0]

3 Jul 2023

Point to point reply for manuscript PONE-D-23-06462 (Body mass index trends and its impact of under and overweight on outcome among PLHIV on antiretroviral treatment in rural Tanzania: A prospective cohort study)

Editor comments

Response: The manuscript formatting guides were followed in the revised manuscript. 

“I have read the journal’s policy and the author of this manuscript has the following competing interests: Emilio Letang is a full-time employee ViiV Healthcare since May 2021. Other authors have declared that no competing interests exist.”

Response: The statement was included in the rebuttal letter as suggested and we can confirm, that there is no change to the PLOS ONE policy on data sharing, as neither the co-Author (Emili Letang) nor ViiV Healthcare has authority over KIULARCO data, which is owned by the Ifakara Health Institute and the Swiss Tropical and Public Health Institute. 

3. One of the noted authors is a group or consortium [KIULARCO study group]. In addition to naming the author group, please list the individual authors and affiliations within this group in the acknowledgments section of your manuscript. Please also indicate clearly a lead author for this group along with a contact email address.

Response: Thank you for the comment. We have updated the KIULARCO study group member list in the acknowledgment section, added the respective affiliations and contacts of the research coordinator of the group.

Response: We have reviewed the reference list to ensure that it is complete and correct. As far as we are aware, none of the cited papers have been retracted

Reviewer #1: 

The manuscript presents a very important topic and is well written, however there are few things which needs to be improved or clarified

1. In the abstract section: BMI categories need to clearly stated eg: the author just mentioned overweight / obese of 25.0kg/m2. this needs to be clear if it is greater than 25 or just 25 kg/m2

 Response: Thank you for the comment. We apologise as the formatting was lost in the version of the abstract that was uploaded to the online system. This appears correctly in the main document that was uploaded with the abstract and main text, which states in the methods section of the abstract in lines #65-66:

Body Mass Index (BMI) was classified as underweight (<18.5kg/m2), normal BMI (18.5–24.9kg/m2), and overweight/obese (≥25.0kg/m2). 

2. Double burden of malnutrition should be clearly defined, and the author should be consistent throughout if it is double burden of malnutrition of just underweight and overweight

Response: Thank you for this comment. We have clarified the definition of the double burden of malnutrition, namely both undernutrition and overnutrition in the same population, in line #81, line #105, lines #336 and 411).

3. Under Table 1, the author should clarify why she/ he included the information that column percentages are presented of non-missing data (this is confusing)

Response: Thank you, column percent has been used (Please see Table 1 and Table S3)

4. Lines 247 -249 the word "their category" need to be specified which category the author is referring

Response: Thank you for the comment. We agree this was unclear and have now improved the sentence (please see line #253-54).

5. Line 254 - thin line and thick lines for figure 1 needs to be more clear, as it is difficult to differentiate which line is thin and which one is thick

Response: The Figure 2 caption is now clarified in lines #259-60.

6. Lines 284 - 285 which results the author is reefing needs be clear

Response: The results being referred to are now clarified in lines #289-90

7. Lines 293 - 294 WHO stages needs to be defined clearly

Response: Clinical stages of HIV infection according to the WHO has been added under methods, lines #154-55 with the respective citation (https://journalofethics.ama-assn.org/article/who-clinical-staging-system-hivaids/2010-03). Please find the detailed definitions below. As these are standardized definitions, we did not add them to the manuscript, but do reference them now 

Stage 1. Patients who are asymptomatic or have persistent generalized lymphadenopathy (lymphadenopathy of at least two sites [not including inguinal] for longer than 6 months) are categorized as being in stage 1, where they may remain for several years. 

Stage 2. Even in early HIV infection, patients may demonstrate several clinical manifestations. Clinical findings included in stage 2 (mildly symptomatic stage) are unexplained weight loss of less than 10 percent of total body weight and recurrent respiratory infections (such as sinusitis, bronchitis, otitis media, and pharyngitis), as well as a range of dermatological conditions including herpes zoster flares, angular cheilitis, recurrent oral ulcerations, papular pruritic eruptions, seborrhoeic dermatitis, and fungal nail infections.

Stage 3. As disease progresses, additional clinical manifestations may appear. Those encompassed by the WHO clinical stage 3 (the moderately symptomatic stage) category are weight loss of greater than 10 percent of total body weight, prolonged (more than 1 month) unexplained diarrhea, pulmonary tuberculosis, and severe systemic bacterial infections including pneumonia, pyelonephritis, empyema, pyomyositis, meningitis, bone and joint infections, and bacteremia. Mucocutaneous conditions, including recurrent oral candidiasis, oral hairy leukoplakia, and acute necrotizing ulcerative stomatitis, gingivitis, or periodontitis, may also occur at this stage. 

Stage 4. The WHO clinical stage 4 (the severely symptomatic stage) designation includes all of the AIDS-defining illnesses. Clinical manifestations for stage 4 disease that allow presumptive diagnosis of AIDS to be made based on clinical findings alone are HIV wasting syndrome, Pneumocystis pneumonia (PCP), recurrent severe or radiological bacterial pneumonia, extrapulmonary tuberculosis, HIV encephalopathy, CNS toxoplasmosis, chronic (more than 1 month) or orolabial herpes simplex infection, esophageal candidiasis, and Kaposi’s sarcoma [4]. Other conditions that should arouse suspicion that a patient is in clinical stage include cytomegaloviral (CMV) infections (CMV retinitis or infection of organs other than the liver, spleen or lymph nodes), extrapulmonary cryptococcosis, disseminated endemic mycoses (e.g., coccidiomycosis, penicilliosis, histoplasmosis), cryptosporidiosis, isosporiasis, disseminated non-tuberculous mycobacteria infection, tracheal, bronchial or pulmonary candida infection, visceral herpes simplex infection, acquired HIV-associated rectal fistula, cerebral or B cell non-Hodgkin lymphoma, progressive multifocal leukoencephalopathy (PML), and HIV-associated cardiomyopathy or nephropathy. Presence of these conditions unaccompanied by the AIDS-defining illnesses, however, should prompt confirmatory testing.

8. Lines 309 It might be important for the author to mention which comorbidities he / she is referring

Response: Thank you. The comorbidities we analysed were tuberculosis and anaemia, however now with the revision done for Table S3, only aneamia was presented (please see line 317). This has been added to the methods section (lines #155-59) and clarified in the results section in lines #315-16

9. Lines 362 - 363 The sentence should be competed, From another study.....which study ??

Response: The sentence has been completed. Please see lines #383-84.

Reviewer #2: 

Thank you very much for the opportunity to read this manuscript. I enjoyed reading it as the overall quality of the study and the writing is excellent. All of my comments following below are suggestions that the authors may want to consider in a revised manuscript.

1. Introduction

Although this is an understudied area, there has been much research on this topic in low-to-middle-income countries over the past couple of years that has not been included in the introduction, although this is indeed largely in urban areas.

Alebel, A., Demant, D., Petrucka, P., & Sibbritt, D. (2021). Effects of undernutrition on mortality and morbidity among adults living with HIV in sub-Saharan Africa: a systematic review and meta-analysis. BMC infectious diseases, 21, 1-20.

Fuseini, H., Gyan, B. A., Kyei, G. B., Heimburger, D. C., & Koethe, J. R. (2021). Undernutrition and HIV infection in sub-Saharan Africa: health outcomes and therapeutic interventions. Current HIV/AIDS Reports, 18, 87-97.

Seid, A., Seid, O., Workineh, Y., Dessie, G., & Bitew, Z. W. (2023). Prevalence of undernutrition and associated factors among adults taking antiretroviral therapy in sub-Saharan Africa: A systematic review and meta-analysis. Plos one, 18(3), e0283502. 

Mahlangu, K., Modjadji, P., & Madiba, S. (2020, August). The nutritional status of adult antiretroviral therapy recipients with a recent HIV diagnosis; a cross-sectional study in primary health facilities in Gauteng, South Africa. In Healthcare (Vol. 8, No. 3, p. 290). MDPI.

Response: Thank you for the valuable reference suggestion, which we have added in the following sections. 

 Alebel et al 2021 and Fuseini H et al 2021: line 112 (introduction)

 Seid A et al 2023: line 349 and 351 (discussion)

 Mahlangu K et al 2020: lines 347 and 362 (discussion)

2. Methods

It would be great to know more about the districts where this data has been collected, particularly considering the limitations of the current body of evidence discussed in the introduction.

Response: Thank you for the comment. We have added further details on the districts, from where patients originate, to the methods section (see lines #127-30). 

3. The authors may also want to comment on the quality of data collection within healthcare services in Tanzania.

Response: We agree that data collection within routine care might not always be optimal. Thanks to the prospective, systematic data capturing system within KIULARCO, we are confident that our BMI assessments were reliable and represent a true picture of reality. This is included as a strength of this study in the discussion. 

4. The statistical analyses have been described well and are appropriate for the outcomes.

Response: Thank you for the positive feedback. 

5. Results

The results are overall well-written. Was an analysis undertaken between those remaining in care and those who where LTFU? Are there differences in demographic characteristics?

Response: Thank you for the comment. The primary outcome of the manuscript was BMI and the secondary outcome was death. We combined death or LTFU because of the knowledge that 40% of traced LTFU patients died. Assessment of the factors associated with LTFU is beyond the scope of this study, and has been previously investigated in this cohort (Please see Ref: Kalinjuma AV, Glass TR, Weisser M, Myeya SJ, Kasuga B, Kisung'a Y, Sikalengo G, Katende A, Battegay M, Vanobberghen F; KIULARCO Study Group. Prospective assessment of loss to follow-up: incidence and associated factors in a cohort of HIV-positive adults in rural Tanzania. J Int AIDS Soc. 2020 Mar;23(3):e25460. doi: 10.1002/jia2.25460. PMID: 32128998; PMCID: PMC7054631.)

Discussion

The discussion is particularly well developed but would benefit from further comparisons with similar studies in other African settings.

Response: Thank you for the comment. We have added the literature as suggested and have added data from other settings for comparison. 

 Seid A et al 2023: line 349 and 351 (discussion)

 Mahlangu K et al 2020: lines 347 and 362 (discussion)

 Further comparisons: lines #359-60, lines #362-63, lines #367-69, lines #376-78, lines #384-89

---

## [Decision Letter · Decision Letter 1]

9 Aug 2023

Body mass index trends and its impact of under and overweight on outcome among PLHIV on antiretroviral treatment in rural Tanzania: A prospective cohort study

PONE-D-23-06462R1

Dear Dr. Kalinjuma,

We’re pleased to inform you that your manuscript has been judged scientifically suitable for publication and will be formally accepted for publication once it meets all outstanding technical requirements.

Kind regards,

I. Marion Sumari-de Boer, Ph.D

Academic Editor

PLOS ONE

Additional Editor Comments (optional):

Reviewers' comments:

Reviewer's Responses to Questions

**Comments to the Author**

1. If the authors have adequately addressed your comments raised in a previous round of review and you feel that this manuscript is now acceptable for publication, you may indicate that here to bypass the “Comments to the Author” section, enter your conflict of interest statement in the “Confidential to Editor” section, and submit your "Accept" recommendation.

Reviewer #1: All comments have been addressed

2. Is the manuscript technically sound, and do the data support the conclusions?

Reviewer #1: Yes

3. Has the statistical analysis been performed appropriately and rigorously? 

Reviewer #1: Yes

4. Have the authors made all data underlying the findings in their manuscript fully available?

Reviewer #1: Yes

5. Is the manuscript presented in an intelligible fashion and written in standard English?

Reviewer #1: Yes

6. Review Comments to the Author

Reviewer #1: The authors have addressed well the comments as raised, and now the paper is more clear and understood

7. PLOS authors have the option to publish the peer review history of their article (what does this mean?). If published, this will include your full peer review and any attached files.

Reviewer #1: **Yes: **Mary Vincent Mosha

---

## [Editor Report · Acceptance letter]

14 Aug 2023

PONE-D-23-06462R1 

Body mass index trends and its impact of under and overweight on outcome among PLHIV on antiretroviral treatment in rural Tanzania: A prospective cohort study 

Dear Dr. Kalinjuma:

I'm pleased to inform you that your manuscript has been deemed suitable for publication in PLOS ONE. Congratulations! Your manuscript is now with our production department. 

Kind regards, 

on behalf of

Dr. I. Marion Sumari-de Boer 

Academic Editor

PLOS ONE